# Aerosol effects on cloud water amounts were successfully simulated by a global cloud-system resolving model

Yousuke Sato [1,2], Daisuke Goto [3], Takuro Michibata [4], Kentaroh Suzuki [5], Toshihiko Takemura [4], Hirofumi Tomita[2] & Teruyuki Nakajima [6]

Aerosols affect climate by modifying cloud properties through their role as cloud condensation nuclei or ice nuclei, called aerosol–cloud interactions. In most global climate models (GCMs), the aerosol–cloud interactions are represented by empirical parameterisations, in which the mass of cloud liquid water (LWP) is assumed to increase monotonically with increasing aerosol loading. Recent satellite observations, however, have yielded contradictory results: LWP can decrease with increasing aerosol loading. This difference implies that GCMs overestimate the aerosol effect, but the reasons for the difference are not obvious. Here, we reproduce satellite-observed LWP responses using a global simulation with explicit representations of cloud microphysics, instead of the parameterisations. Our analyses reveal that the decrease in LWP originates from the response of evaporation and condensation processes to aerosol perturbations, which are not represented in GCMs. The explicit representation of cloud microphysics in global scale modelling reduces the uncertainty of climate prediction.

[1] Department of Applied Energy, Graduate School of Engineering, Nagoya University, Furo-cho, Chikusa-ku, Nagoya, Aichi 464-8603, Japan. [2] RIKEN Advanced Institute for Computational Science, 7-1-26, Minatojima-minami-machi, Chuo-ku, Kobe, Hyogo 650-0047, Japan. [3] National Institute for Environmental Studies, 16-2, Onogawa, Tsukuba, Ibaraki 305-8506, Japan. [4] Research Institute for Applied Mechanics, Kyushu University, 6-1, Kasugakoen, Kasuga, Fukuoka 816-8580, Japan. [5] Atmosphere and Ocean Research Institute, The University of Tokyo, 5-1-5, Kashiwanoha, Kashiwa, Chiba 277-8568, Japan. [6] Earth Observation Research Center, Japan Aerospace Exploration Agency, 2-1-1, Sengen, Tsukuba, Ibaraki 305-8505, Japan. Correspondence and requests for materials should be addressed to Y.S. (email: y-sato@energy.nagoya-u.jp)

Aerosol–cloud interaction (ACI) is thought to exert a significant cooling effect on the Earth's climate, which may offset a substantial fraction of the warming effects of greenhouse gases[1]. However, estimates of the magnitude of this cooling are still uncertain, despite enormous research efforts over the past few decades. In particular, estimation of aerosol-induced modulation of cloudiness, called the lifetime effect[2], still has large uncertainties[1]. In some global climate models (GCMs), this uncertainty has originated from uncertain cloud parameters[3], which tend to be optimised to adjust the magnitude of the aerosol indirect effect so that the models reproduce historical climate changes[4]. Such model tuning is currently being evaluated using recent satellite observations, which provide a constraint on cloud parameters. However, satellite-based model constraints have been found to be inconsistent with the model representations of historical temperature trends as reported in previous studies[4,5]. These studies investigated how climate simulations are sensitive to a particular tunable parameter, the threshold cloud particle radius ($r_{crit}$) for the conversion from cloud to rain, which largely controls the magnitude of the ACI. When a GCM is driven by a value of $r_{crit}$ optimised to reproduce the historical temperature trend, with a tuned magnitude of ACI, the model cannot represent the vertical microphysical structures observed by satellites, and vice versa. This suggests that conventional GCMs contain flawed representations of the aerosol indirect effect, even though they reproduce the historical temperature trend.

A growing body of evidence also suggests that cooling due to the indirect effect may be less than has been estimated by conventional GCMs[6]. One possible cause of this is the lack of climate model representations of the buffered system inherent in the real atmosphere[7], which acts to reduce the cloud response to aerosol perturbations. The buffered system hypothesis is supported by a recent case study of a volcanic eruption[8], which reported a muted response of the liquid water path (LWP) to aerosol perturbations in moderate-resolution imaging spectroradiometer (MODIS) satellite observations, in contrast to GCMs that indicated coherent positive responses of the LWP. This result implies that the lifetime effect can be buffered due to the variation in cloud air mixing, which originates from the aerosol-induced reduction in cloud particle size[9]. These recent findings motivated us to reinvestigate the response of cloud microphysical properties to aerosol perturbations at the global scale.

For the lifetime effect, the cloud response is typically quantified by the aerosol-induced modulation of the cloud LWP, which is given by $\lambda_c = d[LC]/d[CC]$[10]. The notation of each symbol is summarised in Table 1. As in previous studies[10], we targeted only warm-topped clouds, whose cloud top temperatures exceed 273.15 K due to their large contributions to the Earth's energy budget[11]. The traditional theory of the lifetime effect produces positive values of $\lambda_c$ because LWP increases with increasing aerosol loading through inhibition of rain formation due to reduced cloud particle sizes. This is also the major assumption adopted by conventional GCMs, which predict global-mean positive values of $\lambda_c$[10,12,13], with the exception of a few that predict negative values[14,15]. By contrast, $\lambda_c$ estimated from satellite observations can have either positive or negative values depending on the atmospheric stability and precipitation strength[16], with a negative global-mean value[16] (Fig. 1a, d). This discrepancy between model results and satellite observations suggests that conventional GCMs inherently overestimate the magnitude of the lifetime effect.

A recent study reported that this overestimation can be mitigated by using a sophisticated parameterisation of rain[17]. Via a series of the recently developed GCMs, a sophisticated two-moment bulk scheme has been implemented into a GCM[18], and detailed analysis of the overestimation based on the cloud microphysical properties has become possible. However, even using such state-of-the-art GCMs, the issue has not been overcome completely, at least in part because GCMs still rely on horizontal grid resolutions that are too coarse to resolve individual clouds. This resolution issue can be addressed using large-eddy simulation (LES) models that reproduce both negative and positive values of $\lambda_c$ depending on cloud regimes[19]. Another study using an LES model also indicated that the response of LWP to aerosol perturbations can be either positive or negative depending on the relative humidity (RH) above the cloud[20]. These studies using LES models have enabled the uncertainty of the estimation of the lifetime effect to be reduced, but such simulations have been limited to small domains. As a result, there is still a fundamental gap in the understanding of the cloud response to aerosol perturbations between process models and GCMs.

To bridge this gap, we conducted a global simulation using a model that incorporates an explicit representation of cloud microphysical processes[21]. Henceforth, we refer to this simulation as a global cloud system resolving model (GCRM) simulation. Through comparative analyses of the results of the GCRM and GCM, we attempted to understand the reason for the key discrepancy in $\lambda_c$ between GCM results and satellite observations. The analyses elucidated that the GCRM accurately reproduced the negative $\lambda_c$. The negative $\lambda_c$ was originated from the response of the evaporation and condensation processes to the aerosol loading.

## Results

**The GCRM successfully simulated the negative $\lambda_c$.** Figure 1 shows the $\lambda_c$ values estimated from the A-Train satellite and another index of cloud modulation in response to aerosol perturbations, $\lambda = d[LC]/d[AC]$, as a proxy for $\lambda_c$, simulated by the models. Although the absolute value of $\lambda$ is different from that of $\lambda_c$, the signs of $\lambda$ and $\lambda_c$ are similar. The results from our GCRM simulations (Fig. 1b, e) reproduce the regional variability in $\lambda$, including both positive and negative values. The correspondence of $\lambda$ with satellite observations (Fig. 1a, d) was greatly improved with the correct sign of the response being simulated. This is in stark contrast to traditional GCMs that cannot reproduce negative values of $\lambda$ (Fig. 1c, f).

A successful simulation of $\lambda$ by the GCRM that is consistent with observations would offer an unprecedented opportunity to

**Table 1 Summary of the notation and definition of symbols**

| Name of symbols | Definition or meaning |
| --- | --- |
| $N_c$ | Column cloud number concentration |
| $N_a$ | Column aerosol number concentration |
| LWC | Liquid water content (mass of liquid water in each layer) |
| LWP | Liquid water path (vertically integrated LWC) |
| $N_{a,l}$ | Aerosol number concentration in each layer |
| $P_{condensation}$ | Condensation tendency |
| $P_{evaporation}$ | Evaporation tendency |
| CC | $\mathrm{Log}_{10}(N_c)$ |
| AC | $\mathrm{Log}_{10}(N_a)$ |
| LC | $\mathrm{Log}_{10}(\mathrm{LWP})$ |
| $L$ | $\mathrm{Log}_{10}(\mathrm{LWC})$ |
| $A$ | $\mathrm{Log}_{10}(N_{a,l})$ |
| $\lambda_c$ | $d[LC]/d[CC]$ |
| $\lambda$ | $d[LC]/d[AC]$ |
| $\lambda_{lwc}$ | $d[L]/d[A]$ |
| $\lambda_{cond}$ | $d[P_{condensation}]/d[A]$ |
| $\lambda_{evap}$ | $d[P_{evaporation}]/d[A]$ |

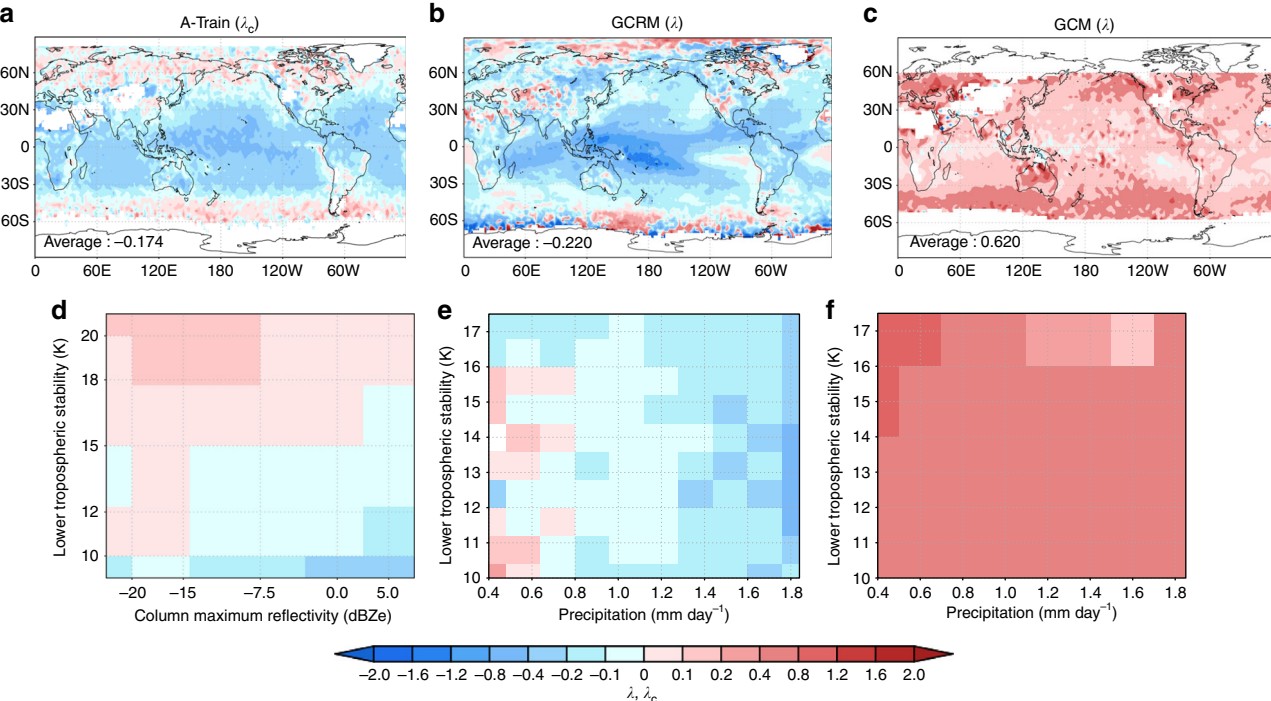

**Fig. 1** Comparison of the response of the liquid water path to the perturbation of the aerosol number concentration estimated from satellite observations and the models. Geographical distribution of **a** $\lambda_c$ and **b**, **c** $\lambda$, and distribution of **d** $\lambda_c$ and **e**, **f** $\lambda$ over the precipitation-lower tropospheric stability (LTS) plane estimated from the results of **a**, **d** the A-Train satellite, **b**, **e** the GCRM, and **c**, **f** the GCM[16]. Positive and negative values of $\lambda$ mean an increase or decrease in LWP with increasing aerosol loading, respectively. For **d**–**f**, the results over the ocean between 60° S to 60° N were used. $\lambda$ in each grid was calculated by the least-square method. The number written at the bottom left of **a**–**c** was the value of $\lambda_c$ and $\lambda$ averaged from 60° S to 60° N. The negative value of average $\lambda$ was successfully reproduced in the GCRM. Figures were mapped using the Grid Analysis and Display System (GrADS)[44] version 2.1.a1

explore the possible mechanisms through which LWP increases or decreases with increasing aerosol loading at the global scale. We investigated the vertical profiles of $\lambda$, denoted as $\lambda_{lwc} = d[L]/d[A]$, targeting clouds over the ocean between 60° S and 60° N. The GCRM predicted positive and negative values of $\lambda_{lwc}$ in the lower (up to 500 m from the cloud base) and upper (above 500 m from the cloud base) layers of clouds, respectively (Fig. 2a). The vertical switch in the sign of $\lambda_{lwc}$ from positive to negative with height suggests that key mechanisms in the liquid water content (LWC) response to aerosol perturbations vary with height. We demonstrate that the responses of two different processes leading to aerosol perturbations, the condensation and evaporation processes, are responsible for the positive and negative LWC responses, respectively.

Aerosol-induced reduction in cloud particle size makes the condensation and evaporation processes efficient[22] when ambient air is saturated and sub-saturated, respectively. In the lower layer of clouds, condensation becomes more active with increasing aerosol loading (Fig. 2b) because of the moist ambient air (Fig. 2d). On the other hand, evaporation becomes more active with increasing aerosol loading (Fig. 2c) because of the dry ambient air (Fig. 2d). The increase in condensation leads to a positive $\lambda_{lwc}$ value in the lower part of the cloud, while the negative $\lambda_{lwc}$ value in the upper part is due to the increase in evaporation. This dependency of $\lambda_{lwc}$ on surrounding environmental conditions supports a previous satellite observation, where LWP was found to decrease with an increase in the aerosol index under low RH conditions and vice versa[23].

The relative contributions of the two competing processes responsible for the positive and negative responses of cloud water to aerosol perturbations vary with altitude. In the lower parts of

clouds, the increase in LWC through the response of the condensation process is larger than the decrease in LWC through the evaporation process, and vice versa in the upper parts of clouds. This vertical switch in the predominant process results in the vertical variability in $\lambda_{lwc}$ values.

These findings can be summarised as follows. Under unstable conditions, characterised by low values of lower tropospheric stability (LTS)[24] with strong precipitation, the cloud thickness exceeds 1000 m (Fig. 2d) and RH is low due to the weak temperature inversion, while $\lambda_{lwc}$ is negative (Fig. 2a). Because the positive value of $\lambda_{lwc}$ in the lower layer is outweighed by the negative value of $\lambda_{lwc}$ in the upper layer, $\lambda$, the value of $\lambda_{lwc}$ for whole layer, becomes negative. By contrast, over the stable region, characterised by high values of LTS with weak precipitation, the cloud thickness is mostly below 500–1000 m (Fig. 2d) due to the strong temperature inversion[24]. In this region, $\lambda$ becomes positive or close to zero, because the contribution of the positive value of $\lambda_{lwc}$ in the lower layer to the column value ($\lambda$) is larger or equal to that of the negative value of $\lambda_{lwc}$ in the upper layer.

## Discussion

The GCRM results described above indicate that condensation and evaporation efficiencies play a critical role in determining the sign of the LWP response to changes in aerosol loading. In particular, vertical variations in these processes and their responses to aerosol perturbations appear to be key to accurate representation of the lifetime effect. A comparison of the vertical structure simulated by the GCRM (Fig. 2) with that simulated by the GCM indicates the reason for the overestimation of $\lambda$ in GCMs. Here, we investigated the vertical structure (Fig. 3). GCMs

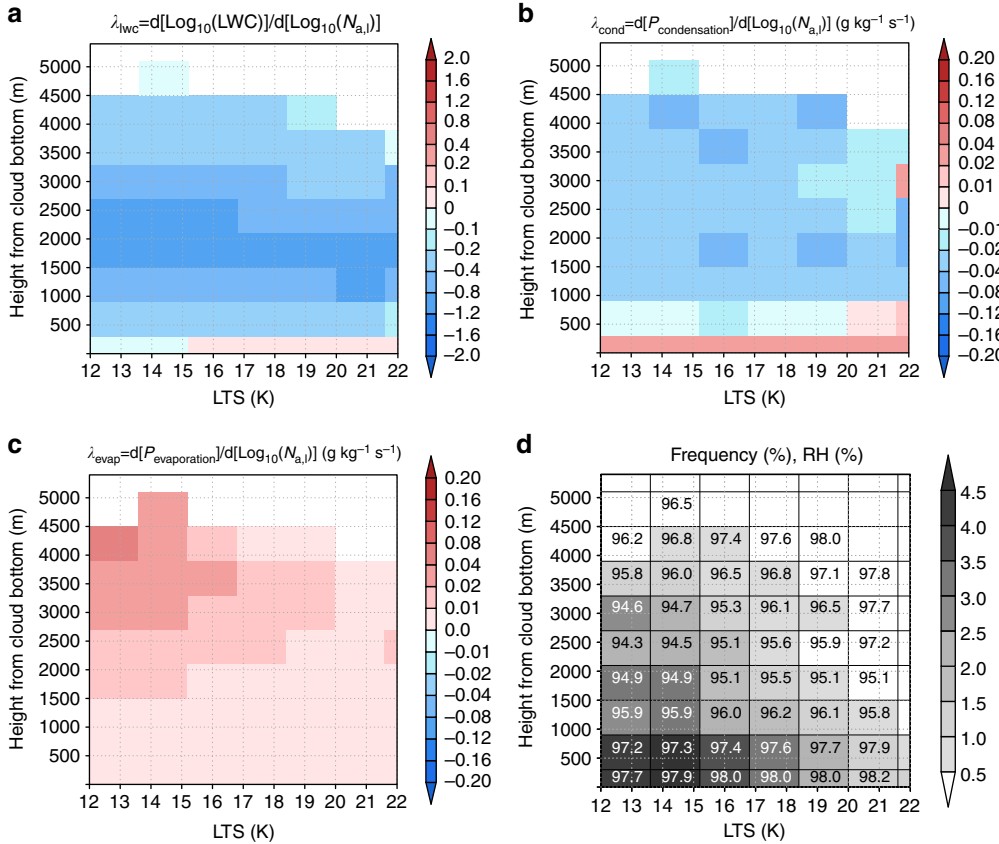

**Fig. 2** The vertical distribution of variables to explain the reasons for the negative value of $\lambda$ simulated by the global cloud system resolving model. Distribution of the **a** $\lambda_{lwc}$, **b** response of the condensation tendency ($P_{condensation}$) to aerosol perturbations ($\lambda_{cond}$), **c** response of the evaporation tendency ($P_{evaporation}$) to aerosol perturbations ($\lambda_{evap}$), and **d** frequency and relative humidity (RH) over the lower tropospheric stability (LTS)-height from the cloud bottom plane simulated by the GCRM. The number in each grid of **d** indicates the RH (%). Results over the ocean between 60° S to 60° N were used for creating these figures. The cloud bottom was determined as the lowest layer of the grid where the average liquid water content (LWC) was larger than $10^{-5}$ kg kg$^{-1}$

cannot reproduce the vertical inhomogeneity in the response of evaporation and condensation processes to aerosol perturbations (Fig. 3b, c). The responses of these processes to aerosol perturbations were much smaller than in the GCRM. Due to this weak response of the evaporation and condensation processes, the negative $\lambda_{lwc}$ in the upper part of the cloud was unclear in the GCM (Fig. 3a). This is because the cloud parameterisation in conventional GCMs does not explicitly include the interaction between the cloud microphysics and dynamics. As a result, conventional GCMs cannot reproduce the negative global-mean value of $\lambda$.

The positive $\lambda$ in the GCM mainly originated from the positive $\lambda_{lwc}$ around the cloud bottom because the cloud thickness simulated by the GCM was mostly smaller than 1000 m (Fig. 3d). The positive $\lambda_{lwc}$ around the cloud bottom originated from the pronounced inhibition of the collision process due to the aerosol-induced reduction of cloud particle sizes, whereas the responses of the evaporation process were not large enough to buffer the cloud water enhancement due to the inhibition of the collision process. The evaporation responses should be appropriately represented in the GCM to reproduce the negative $\lambda$.

The responses of the condensation and evaporation efficiencies to aerosol perturbations buffer the cloud responses to aerosol loadings[7]. Our GCRM simulations, with explicit representations of cloud microphysical processes, enabled us to incorporate this buffering effect at the global scale, and thus to represent the lifetime effect better than has been parameterised previously by GCMs. The GCRM results provide a useful insight into

improvement of parameterisations in current GCMs, particularly when the results are analysed in the context of key cloud processes as described above. Such 'process-oriented' analyses, with combined use of a GCRM and satellite observations, are required to make substantial advances in climate modelling by providing more accurate and reliable climate predictions.

The negative global-mean value of $\lambda$ implies that the radiative forcing due to the lifetime effect, which is usually estimated as negative by GCMs[10], can be tuned into slightly positive. This suggests that estimates of the net negative radiative forcing due to the total ACI[1] can also be significantly reduced and its uncertainty range could even include positive values. This strongly encourages us to estimate radiative forcing based on the GCRM and compare it with GCM values as the next step in this work. In addition, the analyses should also be extended to ice clouds in future studies for a more complete understanding of cloud system responses.

## Methods

**Numerical models.** Global-scale simulations were conducted using a GCRM, the Non-hydrostatic Icosahedral Atmospheric Model (NICAM)[25,26], and a GCM, the Model for Interdisciplinary Research on Climate (MIROC)[27] coupled with an aerosol module, the Spectral Radiation-Transport Model for Aerosol Species (SPRINTARS)[28]. Details of the models and model parameters have been described previously[16,21,29]. The horizontal resolution of GCRM used in this study was 14 km. A one-moment bulk microphysical scheme[30] was adopted, but the cumulus parameterisation was not applied[31] in the GCRM. The vapor mixing ratio ($q_v$), and mixing ratios of cloud ($q_c$), rain ($q_r$), ice ($q_i$), snow ($q_s$), and graupel ($q_g$) were predicted. The condensation/evaporation/sublimation, autoconversion, accretion, gravitational settling, freezing, and melting processes were considered in the

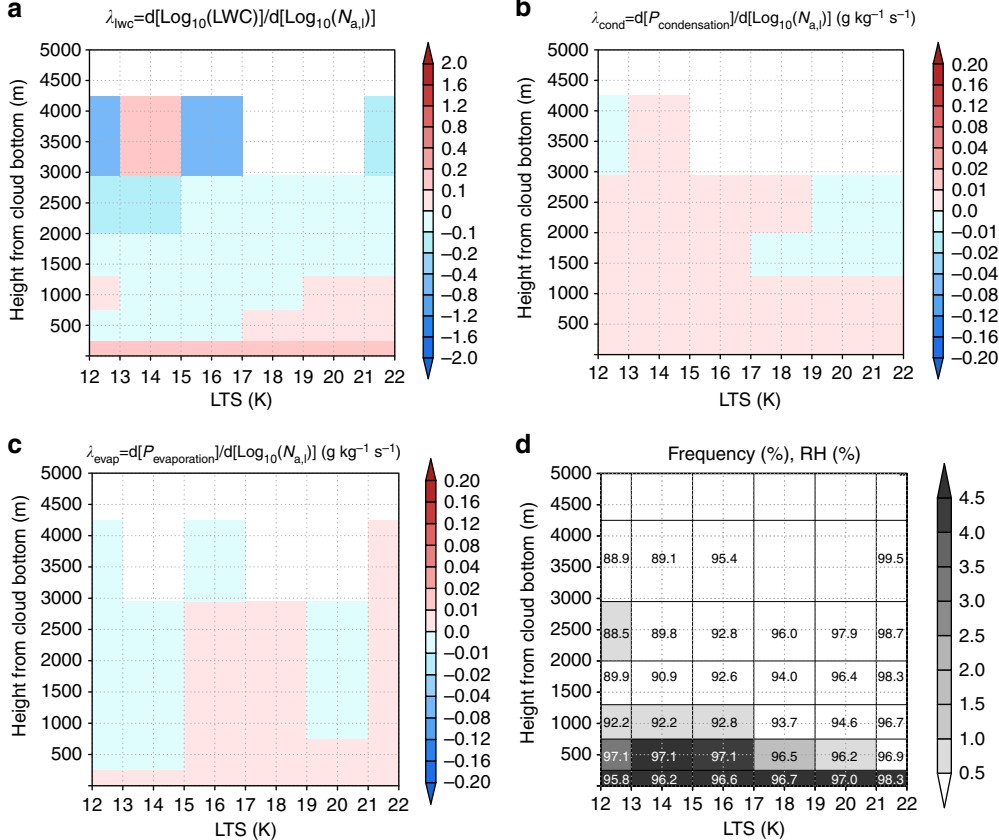

**Fig. 3** The vertical distribution of variables to explain the reasons for the positive value of $\lambda$ by the global climate model. Distribution of the **a** $\lambda_{lwc}$, **b** response of the condensation tendency ($P_{condensation}$) to aerosol perturbations ($\lambda_{cond}$), **c** response of the evaporation tendency ($P_{evaporation}$) to aerosol perturbations ($\lambda_{evap}$), and **d** frequency and relative humidity (RH) over the lower tropospheric stability (LTS)-height from the cloud bottom plane. The number in each grid of **d** indicates the RH (%) simulated by the GCM. The results over the ocean between 60° S to 60° N were used for creating these figures. The cloud bottom was determined as the lowest layer of the grid where the average liquid water content (LWC) was larger than $10^{-5}$ kg kg$^{-1}$

microphysical scheme. Large scale condensation[27] and cumulus parameterisation[32] were applied in the GCM. In the schemes, $q_v$, $q_c$, $q_i$, the cloud droplet number concentration ($N_c$), and the number concentration of ice ($N_i$) were predicted.

In the GRCM, saturation adjustment was applied, and the impacts of the aerosol on cloud microphysical properties were represented by reducing the autoconversion rate with increasing aerosol number concentration as follows. The equation of the autoconversion rate ($P_{auto}$) of the Berry-type scheme[33], which was adopted in both models, incorporates $N_c$ as:

$$P_{auto} = \frac{1}{\rho}\left[a(\rho q_c)^2\left(b + \frac{cN_c}{D_d\rho q_c}\right)^{-1}\right] \qquad (1)$$

where $\rho$ is total density, and $D_d = 0.146 - 5.964 \times 10^{-2} \times \ln(dN_c)$, $a = 16.7$, $b = 5.0$, $c = 3.6 \times 10^{-5}$, $d = 5 \times 10^{-4}$, respectively. The unit of $N_c$ is cm$^{-3}$. The number concentration of cloud condensation nuclei predicted by SPRINTARS was used as $N_c$ in Eq. (1) in GCRM. In the GCM, $N_c$ was predicted by itself and it was used as $N_c$ in Eq. (1).

As several studies pointed out[34–37], there is room for improvement on the one-moment bulk microphysical model. For example, the evaporation of rain simulated by one moment scheme is overestimated[34,36], and the formation of the rain in one moment scheme is faster than that in the sophisticated scheme like two-moment bulk scheme and spectral bin microphysical scheme[35,37]. The global scale simulation by GCRM coupled with such sophisticated scheme should be conducted in future for more accurate and sophisticated discussion about the ACI.

**Experimental setup**. Numerical integrations were conducted for 12 months and 5 years, with time steps ($\Delta t$) of 60 s and 1200 s for the GCRM and GCM after spin-up periods of 1 month and 1 year, respectively. The GCM results for each 1-year period exhibited the same trend as the 5-year simulation, but we used the result of the 5-year simulation to obtain a statistically meaningful value of $\lambda$. For both models, the same series of emission inventories were used. The Hemispheric Transport of Air Pollution Phase 2 (HTAP_v2.2)[38] was used for emission inventories of black carbon, organic carbon, and sulphur dioxide (SO$_2$) from anthropogenic sources. The Global Fire Emissions Database, version 3 (GFEDv3)[39,40] was

used for emissions from biomass burning, and the Global Emissions InitiAtive (GEIA)[41] was used for emissions of terpene and isoprene, which are precursor gases for secondary organic aerosols. The emissions of prescribed monthly oxidants (OH radicals, ozone, and H$_2$O$_2$) were obtained from the results of the GCM coupled with a chemical transport model[42], and the SO$_2$ emitted from volcanoes was obtained from the emission source of a previous study[28].

**Analyses**. The responses of LWP ($\lambda$), LWC ($\lambda_{lwc}$), the condensation tendency ($\lambda_{cond}$), and the evaporation tendency ($\lambda_{evap}$) to aerosol perturbations were calculated by a least-square regression analysis in each 2° × 2° and 2.8° × 2.8° grid of the GCRM and GCM. Grid points with less than five samples were excluded from the analysis.

Only warm-topped clouds whose cloud top temperatures exceeded 273.15 K were analysed due to their large contributions to the Earth's energy budget[11]. A warm-topped cloud was defined as a cloud satisfying the following five conditions: cloud optical thickness > 0.2, LWP > 1 g m$^{-2}$, effective radius at the cloud top > 0.4 μm, cloud top temperature > 273.15 K, and vertically integrated mass of ice water (IWP) < 1 g m$^{-2}$. The cloud top height was defined as the highest layer at which the mass of hydrometeor was over $10^{-5}$ kg kg$^{-1}$, based on a previous study[26].

There is a difference in the data collected by satellites and models that should be noted. Aerosol properties are retrieved from a cloud-free grid in the satellite observations, so cloud and aerosol properties cannot be retrieved simultaneously. By contrast, models can simulate these properties simultaneously for use in analyses. This difference in the data sampling impacts the magnitude of $\lambda$. However, the good agreement of $\lambda$ values determined by the GCRM with satellite observations, particularly the sign, indicates that the microphysical processes, which contribute to the creation of the geographical distribution of $\lambda$ retrieved from satellites, were successfully simulated by the GCRM, but were not simulated by the GCM. Thus, our consideration of the difference between the GCM and the GCRM provides useful information.

The difference in the horizontal axes of Fig. 1d, e, f should also be noted. The horizontal axis of Fig. 1d is the column maximum reflectivity, whereas those of Fig. 1e, f are the precipitation intensity. Given that the column maximum reflectivity is a good indicator of precipitation intensity, based on a previous observational study[43], the meaning of the horizontal axes of Fig. 1d, e, f can be

considered equivalent for the purpose of interpreting the cloud response to aerosol perturbations in the context of precipitation processes.

**Code availability**. The code for NICAM coupled with SPRINTARS associated with this study is available from corresponding author (Y.S.), upon reasonable request. The source code of MIROC coupled with SPRINTARS, associated with this study is available to those who conduct collaborative research with the model users under license from copyright holders. For further information on how to obtain the code please contact the corresponding author (Y.S.).

**Data availability**. The data that support the findings of this study are deposited in local storage at RIKEN/AICS. It is available from the corresponding author (Y.S.) upon reasonable request.

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

## Acknowledgements

Part of the results are obtained by the K computer at the RIKEN Advanced Institute for Computational Science (Proposal number hp150156, hp160004, hp160231, hp170017, hp170232). Simulations of MIROC–SPRINTARS were executed with the SX-ACE supercomputer system of the National Institute for Environmental Studies, Japan. Aspects of this study were supported by the Environment Research and Technology Development Fund (S-12) of the Environmental Restoration and Conservation Agency, by advancement of meteorological and global environmental predictions utilizing observational 'Big Data' of the social and scientific priority issues (Theme 4) to be tackled by using post K computer of the FLAGSHIP2020 Project, by Integrated Research Program for Advancing Climate Models of MEXT, and by the Collaborative Research Program of Research Institute for Applied Mechanics, Kyushu University. The first author was supported by RIKEN special postdoctoral researcher program, and the JSPS Grant-in-Aid for Young Scientists (B) (Grant number: 15K17766). K.S. is supported by NOAA's Climate Program Office's Modeling, Analysis, Predictions and Projections program with grant number NA15OAR4310153. T.T. is supported by Grant-in-Aid for Scientific Research (A) (Grant Number: 15H01728).

## Author contributions

Y.S. designed and analyzed the results of the simulations, and drafted the manuscript. D.G. operated the simulation of NICAM and conducted huge amount of sensitivity experiments using NICAM. T.M. operated the simulation of MIROC. K.S. coordinated the analyses of the simulation and drafted the manuscript. T.T. coordinated the model configuration about SPRINTARS and prepared emission inventory data required for the experiments in this study. H.T. made large amount of discussion about the analyses of the simulation. T.N. submitted the proposal to the fund/computational resources and discussed the design of the numerical experiments, analyses, and scientific knowledge. All co-authors participated in discussions over the results and commented on the original manuscript.

## Additional information

**Competing interests:** The authors declare no competing interests.

