## [Peer Review File · Nature Communications]

Editorial Note: Parts of this peer review file have been redacted as indicated to remove third-party material where no permission to publish could be obtained.

Editorial Note: This manuscript has been previously reviewed at another journal that is not operating a transparent peer review scheme. This document only contains reviewer comments and rebuttal letters for versions considered at Nature Communications. Mentions of prior referee reports have been redacted.

Reviewers' comments:

Reviewer #1 (Remarks to the Author):

This is an important study, still somewhat rough around its edges, the ideas merit publication. The more the edges can be polished the more accessible will be the manuscripts main ideas.

The main accomplishment of this paper is to begin exploring aerosol cloud interactions in models where cloud dynamical processes are beginning to be resolved, so that aerosol cloud interactions then represent an interaction between the resolved flow and the cloud microphysical processes. This is a step advance relative to traditional approaches and something very few groups are capable of. In making this step the authors show that a model which does not prescribe the manner of ACI (as conventional GCMs do) a cloud-mixing negative lifetime effect becomes apparent, in accord with some recent observational and theoretical work, but never before having been seen in a global model. As alluded to in my summary statement the manuscript is rough around the edges, and this should be addressed (following the comments below) before publication. Here the engagement of the senior authors would be beneficial.

1. The manuscript could make better contact with the existing literature. For instance, the seminal study suggesting that humidity could modify the sense of the lifetime effect is Ackerman et al., (Nature, 2004) and this should be referenced. In addition there are now several studies that suggest a negative life-time effect. Generally these are interpreted as arising because smaller drops increase the mixing efficiency of clouds (Stevens, Nature 2017), and there begins to be additional observations, for instance of the Holuhraun eruption (Malavelle, Nature 2017). Drawing on these studies would be helpful. In this context what is a little puzzling is the vertical stratification of the models response, as it appears that the modulation of precipitation is only important for the shallower clouds, and this overwhelms the evaporation effect. Is this because the deeper clouds precipitate regardless, so the perturbation to the mixing efficiency is the only thing that changes?
2. The simulations should be better described, important here is the grid used in NICAM which is often run at quite different resolutions. This does not impact the merit of the results, as whatever the resolution NICAM will under-resolve clouds, but even if it does they will still be better represented than by conventional models. But for subsequent studies, it is important to know more about the details of the simulation, the microphysics model, etc. Also in terms of terminology most people think of CRM as a limited area model and use GCRM for global cloud resolving model. I think this distinction would be helpful and would help distinguish this work from earlier, more idealised, limited area studies.
3. The study would be strengthened if some inference were drawn regarding the contribution of the negative life-time effect to the overall aerosol forcing. Is this commensurate to the expected Twomey effect but of opposite sign (see for instance Seifert, J. Adv. Model. Earth System., 2016) or much smaller?
4. If space were an issue I would prefer more details rather than more caveats. In particular the caveats in the sentence beginning on line 67, and the last sentence of the manuscript add nothing and take away from a stronger presentation of the author's main points.
5. Embedded sentences, wherein a different point is embedded parenthetically, should be avoided. These are fun to write but horrible to read. This should be reformulated so it is not necessary. Sometimes simply saying that that 'the opposite is also true', or 'vice versa' is an easy alternative.
6. Figure 3 is okay in spirit, but should be drafted to avoid having to represent clouds as

Powerpoint dialog boxes turned on their side. Surely after spending a fortune on the simulations a little bit of creative effort could be extended to improving the figures.

7. The formatting of the equations also needs improvement. For instance introducing a mathematical symbol for column Condensate, such as C , then allows the equation (1) to be written simply as $\lambda = \frac{N}{C} \frac{dC}{dN}$. There is no sense in using log-base 10 in this equation. This should also be addressed in the methods section.

8. Some references do not make sense, for instance reference (4) in the summary

Reviewer #2 (Remarks to the Author):

Review of "What is required for accurate global modelling of aerosol effects on cloudiness?" by Sato et al.

This study examined ACI effect for warm clouds from a conventional GCM simulation without an explicit cloud microphysics parameterization and a GCRM simulation, and found that the conventional GCM can not simulate the observed and GCRM-simulated ACI effect because of lacking the physical calculation of condensation and evaporation processes. It is a meaningful study but I do have the following major concerns,

(1) The study was built on the assumption that the conventional GCM does not have an explicit cloud microphysics parameterization. However, nowadays almost all state-of-art GCMs have employed two-moment cloud microphysics parameterization, and cloud liquid is calculated based on microphysical processes such as condensation and evaporation. It is just that the condensation and evaporation are represented crudely based on the saturation adjustment approach in bulk microphysics schemes. So, the study would be more meaningful if the authors propose the problem based on a group of the state-of-art conventional GCMs.

(2) Key results are not shown. The results of Fig 2 should be shown for the conventional GCMs to see how they are different compared with GCRM results and ensure that condensation and evaporation in the conventional GCM are the problematic processes. If the authors say they do not have condensation and evaporation processes in the conventional GCM they used, then the model is out-of-date in the climate field and should not be used for a current study on this topic.

(3) Much key information is missing in methodology. For example, a) what kind of cloud microphysics parameterization is used in the GCM and GCRM? b) For Figure 1, what do you mean by "aerosol variation"? increase or decrease? Are there two different emission scenarios used for the simulations to get the results in Figure 1? If not, explain how do you get the change of LWP to the change of aerosols?

Some specific comments,

1. In the summary, the ACI definition is not right. Aerosol-radiation interactions can modify cloud properties too. ACI is defined as aerosol effects by serving as CCN or IN.

2. The sentence "In global climate models (GCMs), ACI is represented by empirical parameterizations, in which the mass of cloud liquid water (liquid water path: LWP) is assumed to increase monotonically with increasing aerosol loadings across most of the globe" can not be generalized to all GCMs. See my major comment #1. This is only true for the simple GCM used in this study.

3. LTS was not defined when it was first used.

4. Line 114, this sentence is confusing, "the cloud top can distribute beyond 1000 m". The plots show cloud top up to 5000 m.

5. Figure 3. This is not a schematic figure showing anything clearer. The purpose of a schematic figure is to show a mechanism or result in a simpler and easier way than the result plots. I do not think such a figure is needed for a simple physical result shown in this study, which is easy to understand already.

Reviewer #3 (Remarks to the Author):

Summary and Recommendations:

The study introduces a global-scale GCM that represents cloud processes explicitly rather than through empirical parameterizations, to investigate the cloud lifetime or Albrecht effect. Satellite observations from A-train are used as reference and conventional GCMs (with the empirical parameterizations) are shown to produce a response of cloud liquid water path to aerosol that, on the global scale, is opposite in sign to the observations. When the cloud processes are modeled explicitly the correct (relative to obs) sign of the aerosol-LWP response is achieved and the mechanisms for this improvement are examined. The authors find that the effects of changes in cloud drop size due to aerosol on condensation and evaporation processes are responsible, and further that their vertical distributions account for the spatial pattern they manifest over the globe.

This is important and timely work as these processes account for a large amount of uncertainty in the radiative forcing of aerosol-cloud interactions. While I have questions as to how it was specifically done, representing cloud-scale processes in a global-scale model is a large step forward in understanding how to reduce these uncertainties. High resolution models (LES) with smaller domains have produced contradictory results for this problem in a range of different studies.

For this reason I feel this work is important and should be published. Below I have outlined a few comments, some that require some relatively significant clarifications before publication. These may not require significant changes to the manuscript depending on the clarification that the authors are able to provide. ('revise the manuscript' recommendation)

General Comments:

- 1) The authors rely on content in the summary that is not repeated in the body of the text to inform the paper – this is unconventional. I understand there is a space limit, but the body of the paper should stand-alone and provide full understanding of the paper without the summary/abstract. This occurs in the first line (39) with definition of the ACI acronym,
- 2) In several places there is reference to the 'enhancement of cloudiness' by the lifetime effect. But the authors show that aerosol actually reduces LWP on a global average basis, so I would change this language to 'modification of fractional cloudiness' or something similar. This occurs in line 21 of the summary, line 54 in reference to the lifetime/Albrecht effect, and line 78.
- 3) The statement in lines 47-49 is unclear as to what is inconsistent and what is flawed. First, the cloud parameter(s) in question here is (are) very important to specify. Is this only fractional cloudiness? Or LWP? Or something else? When this (these) properties are constrained in the model, if the obs provide truth, what is the mechanism for the incorrect response of the model? (Something other than cloud properties or a different cloud property?) Or is the satellite record of cloud properties in contradiction with the observed temperature record? Clarification is needed here.
- 4) Lines 62-67: Do the authors have information on the differences in how λ is calculated for satellite observations and for models? There is typically much more information available at different scales and distributions in models than in satellite observations. For example, satellites cannot retrieve aerosol properties where there is cloud cover, so neighboring retrievals are used as proxies. Number concentrations and CCN retrievals may be biased by a number of factors. Conversely, models rely on parameterizations and emissions databases to produce fine-scale representations of aerosol and cloud property distribution from which detailed calculations are made at finer scales (but the inputs may be flawed.) These considerations alone probably contribute to some portion of the differences between the two. Can this be quantified or at least briefly discussed?
- 5) In line 76 the statement is made that cloud microphysical processes are explicitly represented in the global CRM. While the method section provides some model details, it is not stated what cloud processes are resolved and how? To what level of detail? While it is interesting on its own to see the difference in the CRM results versus the GCMs with empirical parameterizations, it's difficult for a reader to evaluate the results with any depth without knowing something about the model processes.

6) Following the concern for how lambda is calculated from the models and observations (above # 4) it is stated in line 79 that the CCN concentration from Eq 1 is replaced with total aerosol concentration for the CRM calculations. This must impact the values of lambda if all aerosol are considered to act as CCN? This would increase numbers greatly as it would include many small particles that would not actually activate to CCN. This point definitely needs clarification for publication.

7) In line 82 the claim is made that the CRM and observations agree 'remarkably well.' While I agree that there is a very satisfying improvement in the agreement, there is still some inconsistency such as at high LTS and high dBZ/precip that the language should leave room for. Suggest changing this to something like the agreement is greatly improved and emphasize that the correct sign of the response is achieved. On a secondary note, how well would dBZ and the precip variable extracted from the model be expected to compare?

8) The closing to the paper (lines 137-146) read very nicely and sum up the utility of the results from the current study and directions that should be taken in the future. Nice summary and statement of implications of the paper.

Specific Comments (by line number):

1: title is vague for what is meant by cloudiness – recommend changing title to read "What is required of accurate global modelling of aerosol effects on fractional cloudiness?"

21: aerosol-cloud interactions (plural)

23: in 'most' global climate models

39: Aerosol-cloud interactions (ACI) are considered to exert

43: not sure if 'represented' is the right word here – do you mean mitigated? or offset? or corrected?

46: What is meant by 'emerging satellite observations? Does this refer to emergent properties from satellite observations (as in Bender et al. 2011)? Or does this mean that these satellite observations are newly available and this work could not have been done before without them?

48: inconsistent with the model 'representation' of historical temperature trends...

54: For the lifetime effect (reference to Albrecht 1989), the cloud response...

68: using 'a' sophisticated parameterization

85: change to 'The success in representing the negative lambda seen in observation...' or something similar

92: do the units refer to AMSL? At one point there is a reference to height above cloud base and other to altitude. Consistency and explicit reference for each mention of height would be helpful. in Line 108 there is reference to variation with altitude, Is this specifically correct? Or is there variation with height above cloud base?

167-168: Can this information go in the main body of the text – up front? It changes how the reader thinks about the experiment and the physical processes that are being considered. The rest of the details in this paragraph could be left in this section.

175: literature

254: Part of the results 'are' obtained

Fig 1: lambda label on the color bar would be helpful as well as title labels on the top of the plots 'A-train', 'CRM', 'GCM'

Fig 1: d, e, and f are not geographical distributions – reference to what a-f represent need to be reworded for clarity here.

Fig 1 lines 293-294: Again, can this text be inserted into the text of the paper? It's important info when reading the text and how it is interpreted.

Fig 2 lines 304-307: Same comment – move to main body of the text
Figures themselves are nicely presented.

Reply to the Reviewer #1:

The original title of the manuscript: 'What is required for accurate global modelling of aerosol effects on cloudiness?', by Y. Sato et al. submitted to Nature Communications, whose manuscript number is NCOMMS-17-18141-T.

We are grateful for the reviewer's efforts to read our manuscript and give many useful comments. Based on the reviewer's comments, we modified our manuscript.

In addition, we note that the English of the revised manuscript has been checked by at least two professional editors, both native speakers of English. For a certificate, please see "<http://www.textcheck.com/certificate/p93j7V>".

Several parts of the manuscript were modified from the original manuscript based on the comments of the editors who checked the English.

The modified parts are highlighted in red character.

Our answers written in black letters to the reviewer's comments, which are written in blue letters, are shown below. The Lxx means line number "xx"

We are very happy if the reviewer accepts our manuscript for publication in Nature Communications.

General Comment:

This is an important study, still somewhat rough around its edges, the ideas merit publication. The more the edges can be polished the more accessible will be the manuscripts main ideas.

The main accomplishment of this paper is to begin exploring aerosol cloud interactions in models where cloud dynamical processes are beginning to be resolved, so that aerosol cloud interactions then represent an interaction between the resolved flow and the cloud microphysical processes. This is a step advance relative to traditional approaches and something very few groups are capable of. In making this step the authors show that a model which does not prescribe the manner of ACI (as conventional

GCMs do) a cloud-mixing negative lifetime effect becomes apparent, in accord with some recent observational and theoretical work, but never before having been seen in a global model. As alluded to in my summary statement the manuscript is rough around the edges, and this should be addressed (following the comments below) before publication. Here the engagement of the senior authors would be beneficial.

A: We are very grateful for you to understand the meaning of this study. Based on your very useful comments, we modified our manuscript. Our answers to the comments are shown below. We hope that you understand our answer to the comments, and admit our manuscript to publish from Nature communications.

Specific Comment:

1. The manuscript could make better contact with the existing literature. For instance, the seminal study suggesting that humidity could modify the sense of the lifetime effect is Ackerman et al., (Nature, 2004) and this should be referenced. In addition there are now several studies that suggest a negative life-time effect. Generally these are interpreted as arising because smaller drops increase the mixing efficiency of clouds (Stevens, Nature 2017), and there begins to be additional observations, for instance of the Holuhraun eruption (Malavelle, Nature 2017). Drawing on these studies would be helpful.

A: Thank you for your very useful comment. We added the literature in the revised manuscript. Please see L96 (Ackerman et al. 2004: reference number 20), L69 (Stevens 2017: reference number 9), and L64 (Malavelle et al. 2017: reference number 8) in the revised manuscript.

In this context what is a little puzzling is the vertical stratification of the models response, as it appears that the modulation of precipitation is only important for the shallower clouds, and this overwhelms the evaporation effect. Is this because the deeper clouds precipitate regardless, so the perturbation to the mixing efficiency is the only thing that changes?

A: Thank you for your comment. As you pointed out, the deeper clouds tend to precipitate regardless of the aerosol perturbation and the effect of inhibition of rain formation by aerosol tends to be small in deeper cloud region. The mechanism has impacts on the response of LWP to aerosol perturbation to some extent. However, we

focus on another mechanism to explain the regional variability of the λ as shown below and discussed in the manuscript.

As we shown in Fig. 2b, the condensation tendency increased in the lower part of cloud (~ 500m from the cloud bottom). In contrast, the evaporation tendency increased in the upper part of cloud as shown in Fig. 2c. The responses of the processes are both originated from the reduction of cloud particle size with the increase of aerosols. However, the reduction of cloud particles resulted in different story depending on the moisture of ambient air.

In the lower part of cloud, the ambient air was moist (Fig. 2d) and the condensation tended to be promoted, when the cloud particle size is reduced by the increase of aerosol. On the other hand, the dry air in the upper part of clouds (Fig. 2d) promoted the evaporation process when the cloud particle size is reduced by the increase of aerosol amount. The promotion of evaporation in the upper part and that of condensation in the lower part of cloud occurred regardless of the stability.

In the stable region with high LTS, cloud thickness was smaller than that in unstable region with low LTS (shown in Fig. 2d), because the cloud top is capped by the strong inversion of temperature. In this region, cloud mostly distributed in the layer where the promotion of condensation occurs, and therefore, LWC was increased in the most of cloud layer by the promotion of condensation process. This resulted in the increase of LWP with aerosol, i.e. the positive value of λ .

On the other hand, the cloud thickness was able to be thicker in the unstable region due to the weak inversion (shown in Fig. 2d). In the unstable region, cloud distributed not only the layer, where the promotion of condensation occurred, but also the layer, where the promotion of the evaporation occurred with increasing aerosol amount.

Thus, not only the increase of LWC by the promotion of the condensation in the lower part of clouds, but also the decrease of LWC by the promotion of the evaporation in the upper part of clouds occur in the unstable region.

In view of the column (LWP), the increase of mass (LWC) at lower part of clouds was overwhelmed or cancelled by the decrease of mass (LWC) at upper part of clouds. As a result, LWP was decreased with the increase of aerosols, i.e. the negative value of λ .

This mechanism explains the regional variability of λ . The response of the evaporation and condensation process was not reproduced in GCM as shown in Fig. 3 of the revised manuscript.

2. The simulations should be better described, important here is the grid used in NICAM which is often run at quite different resolutions. This does not impact the merit of the results, as whatever the resolution NICAM will under-resolve clouds, but even if it does they will still be better represented than by conventional models. But for subsequent studies, it is important to know more about the details of the simulation, the microphysics model, etc. Also in terms of terminology most people think of CRM as a limited area model and use GCRM for global cloud resolving model. I think this distinction would be helpful and would help distinguish this work from earlier, more idealised, limited area studies.

A2: Thank you for your useful comments. Based on your comments, we added more detailed description of the microphysical model, emission inventory, and experimental setup into the revised manuscript. Please see the “Methods” of the revised manuscript. In addition, the word “CRM” was modified into “GCRM”. Please see the revised manuscript.

3. The study would be strengthened if some inference were drawn regarding the contribution of the negative life-time effect to the overall aerosol forcing. Is this commensurate to the expected Twomey effect but of opposite sign (see for instance Seifert, J. Adv. Model. Earth System., 2016) or much smaller?

A3: Thank you for your useful comment. As you pointed out, the estimation of the radiative forcing is very important task relating to this work. The negative life-time effect can reduce or cancel the magnitude of positive cloud radiative forcing.

We focus on the response of the cloud to the aerosols in this study, because detail discussions about the response of the cloud to the aerosol perturbation are important as a first step to understand the impact of the radiative forcing. The estimation of the radiative forcing is next step of this study.

Based on your comment, we added some descriptions about the estimation of radiative forcing at the last part of the body of the revised manuscript as:

“The negative global-mean value of λ implies that the radiative forcing due to the lifetime effect, which is usually estimated as negative by GCMs, can be tuned into slightly positive. This suggests that estimates of the net negative radiative forcing due to the total ACI1 can also be significantly reduced and its uncertainty range could even include positive values. This strongly encourages us to estimate radiative forcing based on the GCRM and compare it with GCM values as the next step in this work.”

Please see L192~L197 of the revised manuscript.

4. If space were an issue I would prefer more details rather than more caveats. In particular the caveats in the sentence beginning on line 67, and the last sentence of the manuscript add nothing and take away from a stronger presentation of the author's main points.

A4: Thank you for your useful comment. Based on your comment, we modified the manuscript. We removed the caveats as much as possible, and instead of caveats, we tried to describe in detail. However, some caveats remain in the revised manuscript, based on the comment of other reviewers. I hope the reviewer understand our efforts to modify the manuscript.

5. Embedded sentences, wherein a different point is embedded parenthetically, should be avoided. These are fun to write but horrible to read. This should be reformulated so it is not necessary. Sometimes simply saying that that 'the opposite is also true', or 'vice versa' is an easy alternative.

A5: Thank you for your comments. We removed embedded parenthetically as much as possible in the revised manuscript. Please see the revised manuscript.

6. Figure 3 is okay in spirit, but should be drafted to avoid having to represent clouds as Popwerpoint dialog boxes turned on their side. Surely after spending a fortune on the simulations a little bit of creative effort could be extended to improving the figures.

A: Thank you for your comment. Actually, we have more creative figure than the figure in the first version of the manuscript. However, we removed this figure from the revised manuscript based on the comment of another reviewer.

7. The formatting of the equations also needs improvement. For instance introducing a mathematical symbol for column Condensate, such as C , then allows the equation (1) to be written simply as $\lambda = \frac{N}{C} \frac{\mathrm{d} C}{\mathrm{d} N}$. There is no sense in using log-base 10 in this equation. This should also be addressed in the methods section.

A7: Thank you for your comment. We simplified the equations and added Table 1, which summarizes the notation of symbol, in the revised manuscript. Please see Table 1 of the revised manuscript.

8. Some references do not make sense, for instance reference (4) in the summary

A8: Thank you for your suggestion. Based on your suggestion, we removed the reference (4). Please see the list of the references of the revised manuscript.

Reply to the Reviewer #2:

The original title of the manuscript: 'What is required for accurate global modelling of aerosol effects on cloudiness?', by Y. Sato et al. submitted to Nature Communications, whose manuscript number is NCOMMS-17-18141-T.

We are grateful for the reviewer's efforts to read our manuscript and give many useful comments. Based on the reviewer's comments, we modified our manuscript.

In addition, we note that the English of the revised manuscript has been checked by at least two professional editors, both native speakers of English. For a certificate, please see "<http://www.textcheck.com/certificate/p93j7V>".

Several parts of the revised manuscript were modified from the original manuscript based on the comments of the editors who checked the English.

The modified parts are highlighted in red character.

Our answers written in black letters to the reviewer's comments, which are written in blue letters, are shown below. The Lxx means line number "xx"

We are very happy if the reviewer accepts our manuscript for publication from Nature Communications.

General Comment:

This study examined ACI effect for warm clouds from a conventional GCM simulation without an explicit cloud microphysics parameterization and a GCRM simulation, and found that the conventional GCM can not simulate the observed and GCRM-simulated ACI effect because of lacking the physical calculation of condensation and evaporation processes. It is a meaningful study but I do have the following major concerns,

A: We are very grateful for you to understand the meaning of this study. We answer your comments shown below. We hope you understand our answer and accept our manuscript for publication from Nature communications.

Firstly, we should note that the response of the LWP, LWC, tendencies of evaporation and condensation processes to aerosol perturbations, which are main targets of this

manuscript, were estimated by the least-square regression analysis. As you pointed out in the comment (3)-b), we can estimate the response of them to aerosol perturbations more sophisticatedly by taking the difference between the results of a simulation with present-days (PD) emission scenarios and those of a simulation with pre-industrial (PI) emission scenarios as several previous studies did (e.g., Ghan et al. 2016, Wang et al. 2012, Zhang et al. 2016). However, such estimation (PD-PI) is impossible to be applied for the satellite observations. Thus, we calculated the variation of LWP to the change of aerosols (λ) by using the regression analysis (least-square method) for each grid as several previous studies (Sekiguchi et al. 2003, Michibata et al. 2016).

(1) The study was built on the assumption that the conventional GCM does not have an explicit cloud microphysics parameterization. However, nowadays almost all state-of-art GCMs have employed two-moment cloud microphysics parameterization, and cloud liquid is calculated based on microphysical processes such as condensation and evaporation. It is just that the condensation and evaporation are represented crudely based on the saturation adjustment approach in bulk microphysics schemes. So, the study would be more meaningful if the authors propose the problem based on a group of the state-of-art conventional GCMs.

A(1): Thank you for your useful comment. As you pointed out, the evaporation and condensation are crudely represented in the state-of-the-art GCMs. The GCM used in this study calculated evaporation and condensation processes crudely by the saturation adjustment (Please see “Methods” of the revised manuscript). Even if we use such state-of-the-art GCMs, the problem that clouds are not resolved explicitly in the GCMs due to the coarse spatiotemporal resolution, has not completely resolved. As a result of this problem, the response of the evaporation/condensation process to the aerosol perturbation was not represented in the GCMs.

Based on your comment, we added some descriptions about the state-of-the-art GCMs in the revised manuscript as:

“A recent study reported that this overestimation can be mitigated by using a sophisticated parameterisation of rain. Via a series of the recently developed GCMs, a sophisticated two-moment bulk scheme has been implemented into a GCM, and detailed analysis of the overestimation based on the cloud microphysical properties has

become possible. However, even using such state-of-the-art GCMs, the issue has not been overcome completely, at least in part because GCMs still rely on horizontal grid resolutions that are too coarse to resolve individual clouds.”

Please see L86-L92 and discussion with Fig. 3 of the revised manuscript.

(2) Key results are not shown. The results of Fig 2 should be shown for the conventional GCMs to see how they are different compared with GRCM results and ensure that condensation and evaporation in the conventional GCM are the problematic processes. If the authors say they do not have condensation and evaporation processes in the conventional GCM they used, then the model is out-of-date in the climate field and should not be used for a current study on this topic.

A(2): Thank you for your very useful comment. Based on your comment, we calculated the tendencies of condensation, evaporation, and λ_{wc} from the results of the GCM. We added them as Fig. 3 of the revised manuscript.

As shown in the revised Fig.3, the response of the evaporation and condensation tendency to aerosol perturbations in the GCM are much smaller than that simulated by the GCRM (the λ_{evap} and λ_{cond} estimated by the GCM are about two orders smaller than those of the GCRM). These results mean that the response of the evaporation/condensation process to the aerosol perturbations in the GCM is much smaller than that in the GCRM. As a result of this weak response, the a negative λ_{wc} , which is critical for the negative λ , is not clear in the GCM. (The value of λ_{wc} by the GCM is negative in some parts of upper layer of clouds (Fig. 3a), but the frequency of the cloud in these parts is small.)

In addition, in the lower part cloud, λ_{wc} shows positive, which is originated from the increase of LWC due to the slowdown of autoconversion speed with increasing aerosol. From these results, we can conclude that globally positive λ in the GCM is created by the slowdown of autoconversion speed. And negative value of λ cannot reproduce in the GCM because GCM cannot reproduce the response of condensation/evaporation process in upper part of cloud.

We added the description about the difference in the tendency of condensation/evaporation between the GCM and the GCRM in the revised manuscript. Please see “Discussion” of the revised manuscript.

(3) Much key information is missing in methodology.

A(3): Thank you for your comment. Based on your comment and same comment from other reviewers, we added details of model description in “Methods” section of the revised manuscript. Please see “Methods” of the revised manuscript.

For example, a) what kind of cloud microphysics parameterization is used in the GCM and GCRM?

A(3)-a): Based on your comment, we added more detailed description of the both models. Please see “Methods” of the revised manuscript.

b) For Figure 1, what do you mean by “aerosol variation”? increase or decrease? Are there two different emission scenarios used for the simulations to get the results in Figure 1? If not, explain how do you get the change of LWP to the change of aerosols?

A(3) b): The word “aerosol variation” includes the increase or decrease of aerosol.

As we described in the answer to the general comment, only a simulation with present-days (PD) emission scenarios was conducted in this study. The λ was estimated from the regression analyses in this study. This is because the estimation of λ by taking the difference between PI and PD (PD-PI) condition is impossible for the satellite observation.

As you pointed out, the word “aerosol variation” would make readers infer that the λ was estimated by PD-PI analyses. Thus, we use “aerosol perturbation” instead of “aerosol variation” in the revised manuscript based on your comment. Please see caption of Fig 1 in the revised manuscript. In addition, we modified the word “least-square method” to “least-square regression analyses” in the revised manuscript. Please see L245 of the revised manuscript.

Some specific comments.

1. In the summary, the ACI definition is not right. Aerosol-radiation interactions can modify cloud properties too. ACI is defined as aerosol effects by serving as CCN or IN.

A1: Thank you for your comment. Based on your comment, we modified first sentence of the summary as “Aerosols affect climate by modifying cloud properties through their role as cloud condensation nuclei or ice nuclei, called aerosol–cloud interactions (ACIs)”.

Please see L26-L27 of the revised manuscript.

2. The sentence “In global climate models (GCMs), ACI is represented by empirical parameterizations, in which the mass of cloud liquid water (liquid water path: LWP) is assumed to increase monotonically with increasing aerosol loadings across most of the globe” can not be generalized to all GCMs. See my major comment #1. This is only true for the simple GCM used in this study.

A2: Thank you for your comment. Firstly, we note that we modified “In global climate models” to “In most global climate models” in the revised manuscript based on your comment and that of another reviser. Please see L29 of the revised manuscript.

As you pointed out and as we discussed in the answer for comment 1 (A1) shown above, the response of the microphysical process is crudely reproduced in the state-of-the-art GCMs (including our GCM). However, even if the state-of-the-art GCMs are used, the global mean value of λ_c is positive as shown in an previous intercomparison study (AeroCOM; Ghan et al. 2016). This result means that most of the GCMs including the state-of-the-art GCMs overestimate the λ , and the reason of the overestimation should be investigated.

The discussions of this study with Fig. 3 elucidate that the response of the evaporation/condensation process to aerosol perturbation was too weak to reproduce the negative λ .

Please see the discussions with Fig. 3 of the revised manuscript.

3. LTS was not defined when it was first used.

A3: Thank you for your detailed check. We added full description of LTS, when it was firstly used (caption of Fig 1). Please see the caption of Fig 1 in the revised manuscript.

4. Line 114, this sentence is confusing, “the cloud top can distribute beyond 1000 m”. The plots show cloud top up to 5000 m.

A4: Thank you for your comment. As you pointed out, the sentence “the cloud top can distribute beyond 1000 m” would make reader misunderstand. This sentence should be “the cloud thickness exceeds 1000m”. We modified this sentence in the revised manuscript. Please see L148-L149 of the revised manuscript.

5. Figure 3. This is not a schematic figure showing anything clearer. The purpose of a schematic figure is to show a mechanism or result in a simpler and easier way than the result plots. I do not think such a figure is needed for a simple physical result shown in this study, which is easy to understand already.

A5: Thank you for your comment. Based on your comment, we removed the figure from the revised manuscript.

Reference

Gettelman, A., and H. Morrison, 2015: Advanced two-moment bulk microphysics for global models. part I: Off-line tests and comparison with other schemes. *J. Clim.*, **28**, 1268–1287, doi:10.1175/JCLI-D-14-00102.1.

<http://journals.ametsoc.org/doi/abs/10.1175/JCLI-D-14-00102.1>.

Ghan, S., and Coauthors, 2016: Challenges in constraining anthropogenic aerosol effects on cloud radiative forcing using present-day spatiotemporal variability.

Proc. Natl. Acad. Sci., 201514036, doi:10.1073/pnas.1514036113.

<http://www.pnas.org/lookup/doi/10.1073/pnas.1514036113>.

Michibata, T., K. Suzuki, Y. Sato, and T. Takemura, 2016: The source of discrepancies in aerosol–cloud–precipitation interactions between GCM and A-Train retrievals.

Atmos. Chem. Phys., **16**, 15413–15424, doi:10.5194/acp-16-15413-2016.

<http://www.atmos-chem-phys-discuss.net/acp-2016-831/>.

Sekiguchi, M., T. Nakajima, K. Suzuki, K. Kawamoto, A. Higurashi, D. Rosenfeld, I. Sano, and S. Mukai, 2003: A study of the direct and indirect effects of aerosols

using global satellite data sets of aerosol and cloud parameters. *J. Geophys. Res.*, **108**, 4699, doi:10.1029/2002JD003359.

<http://doi.wiley.com/10.1029/2002JD003359>

Wang, M., and Coauthors, 2012: Constraining cloud lifetime effects of aerosols using A-Train satellite observations. *Geophys. Res. Lett.*, **39**, L15709,

doi:10.1029/2012GL052204. <http://doi.wiley.com/10.1029/2012GL052204>.

Zhang, S., and Coauthors, 2016: On the characteristics of aerosol indirect effect based on dynamic regimes in global climate models. *Atmos. Chem. Phys.*, **16**, 2765–2783,

doi:10.5194/acp-16-2765-2016.

<http://www.atmos-chem-phys-discuss.net/15/23683/2015/>.

Reply to the Reviewer #3:

The original title of the manuscript: 'What is required for accurate global modelling of aerosol effects on cloudiness?', by Y. Sato et al. submitted to Nature Communications, whose manuscript number is NCOMMS-17-18141-T.

We are grateful for the reviewer's efforts to read our manuscript and give many useful comments. Based on the reviewer's comments, we modified our manuscripts.

In addition, we note that the English of the revised manuscript has been checked by at least two professional editors, both native speakers of English. For a certificate, please see "<http://www.textcheck.com/certificate/p93j7V>".

Several parts of the revised manuscript were modified from the original manuscript based on the comments of the editors who checked the English.

The modified parts are highlighted in red character.

Our answers written in black letters to the reviewer's comments, which are written in blue letters, are shown below. The Lxx means line number "xx"

We are very happy if the reviewer accepts our manuscript for publication from Nature Communications.

General Comment:

The study introduces a global-scale GCM that represents cloud processes explicitly rather than through empirical parameterizations, to investigate the cloud lifetime or Albrecht effect. Satellite observations from A-train are used as reference and conventional GCMs (with the empirical parameterizations) are shown to produce a response of cloud liquid water path to aerosol that, on the global scale, is opposite in sign to the observations. When the cloud processes are modeled explicitly the correct (relative to obs) sign of the aerosol-LWP response is achieved and the mechanisms for this improvement are examined. The authors find that the effects of changes in cloud drop size due to aerosol on condensation and evaporation processes are responsible, and further that their vertical distributions account for the spatial pattern they manifest over

the globe.

This is important and timely work as these processes account for a large amount of uncertainty in the radiative forcing of aerosol-cloud interactions. While I have questions as to how it was specifically done, representing cloud-scale processes in a global-scale model is a large step forward in understanding how to reduce these uncertainties. High resolution models (LES) with smaller domains have produced contradictory results for this problem in a range of different studies.

For this reason I feel this work is important and should be published. Below I have outlined a few comments, some that require some relatively significant clarifications before publication. These may not require significant changes to the manuscript depending on the clarification that the authors are able to provide. ('revise the manuscript' recommendation)

A: We are very grateful for the reviewer to understand the meaning of this study, and appreciate the meaning our work. Based on your very useful comments, we modified our manuscript. Our answers to the comments are shown below. We hope that the reviewer understands our answer to the comments, and admit our manuscript to publish from Nature communications.

General Comments:

1) The authors rely on content in the summary that is not repeated in the body of the text to inform the paper – this is unconventional. I understand there is a space limit, but the body of the paper should stand-alone and provide full understanding of the paper without the summary/abstract. This occurs in the first line (39) with definition of the ACI acronym,

A1): Thank you for your comment. Based on your comment, we added full description of abbreviations defined in summary (i.e., ACI, LWP, and GCM) into the body of the manuscript too. In addition, we modified summary to comply with the format of Nature Communications. Please see the summary and L41, L65 and L46 of the revised manuscript.

2) In several places there is reference to the 'enhancement of cloudiness' by the lifetime effect. But the authors show that aerosol actually reduces LWP on a global average basis, so I would change this language to 'modification of fractional cloudiness' or

something similar. This occurs in line 21 of the summary, line 54 in reference to the lifetime/Albrecht effect, and line 78.

A2): Thank you for your comment. Based on your comment, we modified “enhancement of cloudiness” to “modulation of cloudiness”, or similar word. Please see L27, and L73 of the revised manuscript.

3) The statement in lines 47-49 is unclear as to what is inconsistent and what is flawed. First, the cloud parameter(s) in question here is (are) very important to specify. Is this only fractional cloudiness? Or LWP? Or something else? When this (these) properties are constrained in the model, if the obs provide truth, what is the mechanism for the incorrect response of the model? (Something other than cloud properties or a different cloud property?) Or is the satellite record of cloud properties in contradiction with the observed temperature record? Clarification is needed here.

A3): Thank you for your comment. In this sentence, we want to show an example that a tuning parameter constrained with satellite observation cannot reproduce the historical temperature trend. The example of the parameter is threshold value of the cloud radius of autoconversion (r_{crit}), which tunes the magnitude of the ACI in some models. Once the cloud radius exceeds r_{crit} , the conversion from clouds to rain occurs. Previous studies (Golaz et al. 2013; Suzuki et al. 2013), which were cited in the revised manuscript, indicated that result of the GCM tuned by r_{crit} for reproducing historical trend of the temperature couldn't reproduce the cloud microphysical property (vertical structure of radar reflectivity) retrieved from the satellite. On the other hand, GCM tuned by r_{crit} for reproducing the cloud microphysical property couldn't reproduce the historical temperature trend. These results indicate that the “artificial” tuning for reproducing an aspect (historical temperature trend) is not always good for simulating another aspect (microphysical properties).

To show this more clearly, we added more detailed description into the revised manuscript as:

“However, satellite-based model constraints have been found to be inconsistent with the model representations of historical temperature trends as reported in previous studies. These studies investigated how climate simulations are sensitive to a particular tunable

parameter, the threshold cloud particle radius (r_{crit}) for the conversion from cloud to rain, which largely controls the magnitude of the ACI. When a GCM is driven by a value of r_{crit} optimised to reproduce the historical temperature trend, with a ‘tuned’ magnitude of ACI, the model cannot represent the vertical microphysical structures observed by satellites, and vice versa. This suggests that conventional GCMs contain flawed representations of the aerosol indirect effect, even though they reproduce the historical temperature trend.”

Please see L50-L59 of the revised manuscript.

4) Lines 62-67: Do the authors have information on the differences in how lambda is calculated for satellite observations and for models? There is typically much more information available at different scales and distributions in models than in satellite observations. For example, satellites cannot retrieve aerosol properties where there is cloud cover, so neighboring retrievals are used as proxies. Number concentrations and CCN retrievals may be biased by a number of factors. Conversely, models rely on parameterizations and emissions databases to produce fine-scale representations of aerosol and cloud property distribution from which detailed calculations are made at finer scales (but the inputs maybe flawed.) These considerations alone probably contribute to some portion of the differences between the two. Can this be quantified or at least briefly discussed?

A4): Thank you for your comment. As you pointed out, the satellite cannot retrieve cloud and aerosol in the same grid simultaneously. In contrast, we can use both data of aerosol and cloud in the same grid point simultaneously from the model results. In addition, the frequency of data and spatial resolution of the satellite observation are different from those of the model. Such differences in data sampling have some impacts on the value of λ .

Even though the effects of the sampling issue are considered, the geographical distribution of λ estimated from A-Train satellite and GCRM is so similar (as shown in Fig. 1) that we cannot explain the similarity by accidental. So, we propose that the difference between λ estimated from the GCM and that from GCRM obtained by this study is meaningful.

As Nakajima and Schulz (2009) summarized (Fig. A1), several previous studies have estimated λ or λ_c from numerical GCMs and satellite observations (e.g., Sekiguchi et al. 2003, Suzuki et al. 2004, Quaas et al. 2004, Kaufman et al. 2005, Nakajima and Schulz 2009, Michibata et al. 2016). If the difference in the sign of λ (main topic of this study) is only originated from the effects of the sampling issue, the value of λ estimated from all models show positive and the value of λ estimated from all satellite observation show negative. However, the value of λ estimated from models and observations has large variability (both positive and negative). Thus, we think that the variability in the sign of λ does not only come from the sampling issue.

Idealistically, we should estimate the impact of the sampling issues on the value of λ . However, it is difficult to estimate, because there are many other elements to contribute to the variability of the value of λ , (e.g., uncertainties of numerical models, bias of observations, data sampling, and so on).

We agree with your comment, and therefore, we added some descriptions about the difference in the data sampling as following:

“There is a difference in the data collected by satellites and models that should be noted. Aerosol properties are retrieved from a cloud-free grid in the satellite observations, so cloud and aerosol properties cannot be retrieved simultaneously. By contrast, models can simulate these properties simultaneously for use in analyses. This difference in the data sampling impacts the magnitude of λ . However, the good agreement of λ values determined by the GCRM with satellite observations, particularly the sign, indicates that the microphysical processes, which contribute to the creation of the geographical distribution of λ retrieved from satellites, were successfully simulated by the GCRM, but were not simulated by the GCM. Thus, our consideration of the difference between the GCM and the GCRM provides useful information.”

in the revised manuscript based on your comment. Please see L254-L263 of the revised manuscript.

Redacted figure

5) In line 76 the statement is made that cloud microphysical processes are explicitly represented in the global CRM. While the method section provides some model details, it is not stated what cloud processes are resolved and how? To what level of detail? While it is interesting on it's own to see the difference in the CRM results versus the GCMs with empirical parameterizations, it's difficult for a reader to evaluate the results with any depth without knowing something about the model processes.

A5): Thank you for your comment. Based on your comment and the same comment from other reviewers, we added details of model description in "Methods" of the revised manuscript. Please see "Methods" of the revised manuscript.

6) Following the concern for how lambda is calculated from the models and observations (above # 4) it is stated in line 79 that the CCN concentration from Eq 1 is replaced with total aerosol concentration for the CRM calculations. This must impact the values of lambda if all aerosol are considered to act as CCN? This would increase

numbers greatly as it would include many small particles that would not actually activate to CCN. This point definitely needs clarification for publication.

A6): Thank you for your comments. As you pointed out, the absolute value of λ is different from that of λ_c , and one of the reason of the difference is effects of small aerosol particles. However, the sign of λ and λ_c is same even though the absolute value of is different from each other. Of course, we confirmed that the geophysical distribution of the sign of λ_c is similar to the sign of λ . One of the important point of this paper is that “the geophysical distribution of the negative λ can be reproduced by using GCRM”.

We agree your comment, and we added some descriptions about the difference between the λ and λ_c in the revised manuscript as:

“Although the absolute value of λ is different from that of λ_c , the signs of λ and λ_c are similar”

Please see L111-L112 of the revised manuscript.

7) In line 82 the claim is made that the CRM and observations agree ‘remarkably well.’ While I agree that there is a very satisfying improvement in the agreement, there is still some inconsistency such as at high LTS and high dBZ/precip that the language should leave room for. Suggest changing this to something like the agreement is greatly improved and emphasize that the correct sign of the response is achieved.

A7)-1: Thank you for your suggestion. Based on your suggestion, we firstly modified this part into “the agreement is greatly improved and emphasize that the correct sign of the response is achieved”. However, the editors who checked our English suggest another expression of this sentence as:

“The correspondence of λ with satellite observations (Figs. 1a and 1d) was greatly improved with the correct sign of the response being simulated.”

So, we finally adopted this sentence. Please see L114-L115 of the revised manuscript.

On a secondary note, how well would dBZ and the precip variable extracted from the model be expected to compare?

A7)-2: Thank you for your question. We understand that dBZ and precipitation amount is different, but the maximum reflectivity has good indicator of the precipitation intensity (Comstock et al. 2004). The important point shown by Fig 1d-1f is that λ is negative under the unstable (low LTS) condition with strong precipitation, and positive under the stable (high LTS) condition with week precipitation condition. In this sense, the important point is not changed even if the horizontal axis of Fig 1d (dBZ) and Figs 1e and 1f (precipitation) is different. We understand your comment, and we added the sentence

“The difference in the horizontal axes of Figs 1d, 1e, and 1f should also be noted. The horizontal axis of Fig. 1d is the column maximum reflectivity, whereas those of Figs 1e and 1f are the precipitation intensity. Given that the column maximum reflectivity is a good indicator of precipitation intensity, based on a previous observational study, the meaning of the horizontal axes of Figs 1d, 1e, and 1f can be considered equivalent for the purpose of interpreting the cloud response to aerosol perturbations in the context of precipitation processes. ”

in the “Methods” of the revised manuscript. Please see L264-L270 of the revised manuscript.

8) The closing to the paper (lines 137-146) read very nicely and sum up the utility of the results from the current study and directions that should be taken in the future. Nice summary and statement of implications of the paper.

A8): Thank you for your useful comment. Based on your comment, we added description about future study (estimation of radiative forcing) at the last part of the body of the revised manuscript.

Please see L192-L199 of the revised manuscript.

Specific Comments (by line number):

1: title is vague for what is meant by cloudiness – recommend changing title to read “What is required of accurate global modelling of aerosol effects on fractional cloudiness?”

A: Thank you for your suggestion. As you pointed out, the cloudiness not suitable word to indicate the main topic of this work: the response of “LWP”. We modified the title as

“What is required of accurate global modelling of aerosol effects on cloud water amounts?”.

Please see the title of the revised manuscript.

21: aerosol-cloud interactions (plural)

A: Thank you for your reading in detail. We modified “aerosol-cloud interaction” to plural form. Please see L27 of the revised manuscript.

23: in ‘most’ global climate models

A: Thank you for detailed reading. We added “most”. Please see L29 of the revised manuscript.

39: Aerosol-cloud interactions (ACI) are considered to exert

A: Thank you for detailed reading. We added “aerosol-cloud interaction” and included “ACI” into a parenthesis. Please see L41 of the revised manuscript.

43: not sure if ‘represented’ is the right word here – do you mean mitigated? or offset? or corrected?

A: Thank you for your comment. We modified this sentence as:
“this uncertainty has originated from uncertain cloud parameters” .

Please see L46-47 of the revised manuscript.

46: What is meant by ‘emerging satellite observations? Does this refer to emergent properties from satellite observations (as in Bender et al. 2011)? Or does this mean that these satellite observations are newly available and this work could not have been done before without them?

A: Thank you for your detailed reading. The word “emerging” is not necessary in the revised manuscript, and therefore we replace the word “emerging” into “recent” in the revised manuscript. Please see L49 of the revised manuscript.

48: inconsistent with the model ‘representation’ of historical temperature trends...

A: Thank you for your suggestion. Based on your suggestion, we changed ‘performance’ into ‘representations’. Please see L51 of the revised manuscript.

54: For the lifetime effect (reference to Albrecht 1989), the cloud response...

A: Thank you for your suggestion. We added “For the lifetime effect” before “the cloud response.....”. However, we did not add the reference to Albrecht (1989) in this part because the reference was already shown before this part. Please see L72 of the revised manuscript.

68: using ‘a’ sophisticated parameterization

A: Thank you for your suggestion. We added “a” into the part. Please see L86-L87 of the revised manuscript.

85: change to ‘The success in representing the negative lambda seen in observation...’ or something similar

A: Thank you for your suggestion. Based on your suggestion, we modified this part to “A successful simulation of λ by the GCRM that is consistent with observations would offer an unprecedented opportunity to explore the possible mechanisms through which LWP increases or decreases with increasing aerosol loading at the global scale”. Please see L118-L121 of the revised manuscript.

92: do the units refer to AMSL? At one point there is a reference to height above cloud base and other to altitude. Consistency and explicit reference for each mention of height would be helpful. in Line 108 there is reference to variation with altitude, Is this specifically correct? Or is there variation with height above cloud base?

A: As it shown in the vertical axis of Fig 2, the units refer to “m” from “the cloud base (cloud bottom)”. Based on your comment, we added “from the cloud base” after the unit. Please see L123 – L124 of the revised manuscript.

167-168: Can this information go in the main body of the text – up front? It changes

how the reader thinks about the experiment and the physical processes that are being considered. The rest of the details in this paragraph could be left in this section.

A: Thank you for your suggestion. Based on your suggestion, we added following sentence in the body of the revised manuscript,

“As in previous studies, we targeted only warm-topped clouds, whose cloud top temperatures exceed 273.15 K due to their large contributions to the Earth’s energy budget”

Please see L74-76 of the revised manuscript.

175: literature

A: Thank you for your suggestion. We modified the sentence with “literatures” into “Details of the models and model parameters have been described previously”. Please see L206-L207 of the revised manuscript.

254: Part of the results ‘are’ obtained

A: Thank you for your suggestion. We modified “is” into “are”. Please see L380 of the revised manuscript.

Fig 1: lambda label on the color bar would be helpful as well as title labels on the top of the plots ‘A-train’, ‘CRM’, ‘GCM’

A: Thank you for your useful comment. We modified Fig. 1 based on your comment. Please see Fig 1 of the revised manuscript.

Fig 1: d, e, and f are not geographical distributions – reference to what a-f represent need to be reworded for clarity here.

A: Thank you for your comment. Based on your comment, we changed caption of Figure 1 into

“Geographical distribution of (a) λ_c and (b, c) λ , and distribution of (d) λ_c and (e, f) λ over the precipitation-lower tropospheric stability (LTS) plane estimated from the results of (a, d) the A-Train satellite, (b, e) the GCRM, and (c, f) the GCM. Positive and

negative values of λ mean an increase or decrease in LWP with increasing aerosol loading, respectively. For (d–f), the results over the ocean between 60° S to 60° N were used. λ in each grid was calculated by the least-square method. The number written at the bottom left of (a–c) was the value of λ_c and λ averaged from 60° S to 60° N. The negative value of average λ was successfully reproduced in the GCRM. Figures were mapped using the Grid Analysis and Display System (GrADS) version 2.1.a1 “

Please see caption of Fig 1 in the revised manuscript.

Fig 1 lines 293-294: Again, can this text be inserted into the text of the paper? It’s important info when reading the text and how it is interpreted.

Fig 2 lines 304-307: Same comment – move to main body of the text
Figures themselves are nicely presented.

A: Thank you for your suggestion. Based on your suggestion, we added a sentence;

“We investigated the vertical profiles of λ , denoted as $\lambda_{wc} = d[L]/d[A]$, targeting clouds over the ocean between 60°S and 60°N.”

in the body of the revised manuscript. Please see L121-L122 of the revised manuscript.

Reference:

- Comstock, K. K., R. Wood, S. E. Yuter, and C. S. Bretherton, 2004: Reflectivity and rain rate in and below drizzling stratocumulus. *Q. J. R. Meteorol. Soc.*, **130**, 2891–2918, doi:10.1256/qj.03.187. <http://doi.wiley.com/10.1256/qj.03.187>
- Golaz, J.-C., L. W. Horowitz, and H. Levy, 2013: Cloud tuning in a coupled climate model: Impact on 20th century warming. *Geophys. Res. Lett.*, **40**, 2246–2251, doi:10.1002/grl.50232. <http://doi.wiley.com/10.1002/grl.50232>.
- Kaufman, Y. J., I. Koren, L. a Remer, D. Rosenfeld, and Y. Rudich, 2005: The effect of smoke, dust, and pollution aerosol on shallow cloud development over the Atlantic Ocean. *Proc. Natl. Acad. Sci. U. S. A.*, **102**, 11207–11212, doi:10.1073/pnas.0505191102.
- Michibata, T., K. Suzuki, Y. Sato, and T. Takemura, 2016: The source of discrepancies in aerosol–cloud–precipitation interactions between GCM and A-Train retrievals.

- Atmos. Chem. Phys.*, **16**, 15413–15424, doi:10.5194/acp-16-15413-2016.
<http://www.atmos-chem-phys-discuss.net/acp-2016-831/>.
- Nakajima, T., and M. Schulz, 2009: What do we know about large-scale changes of aerosols, clouds, and the radiation budget? *Struüingmann Forum Report: Clouds in the Perturbed Climate System. Their Relationship to Energy Balance, Atmospheric Dynamics, and Precipitation*, J.H. Charlson and J. Robert, Eds., MIT Press, Cambridge, p. 597.
- Quaas, J., O. Boucher, and F. M. Bréon, 2004: Aerosol indirect effects in POLDER satellite data and the Laboratoire de Météorologie Dynamique-Zoom (LMDZ) general circulation model. *J. Geophys. Res. D Atmos.*, **109**, 1–9, doi:10.1029/2003JD004317.
- Sekiguchi, M., T. Nakajima, K. Suzuki, K. Kawamoto, A. Higurashi, D. Rosenfeld, I. Sano, and S. Mukai, 2003: A study of the direct and indirect effects of aerosols using global satellite data sets of aerosol and cloud parameters. *J. Geophys. Res.*, **108**, 4699, doi:10.1029/2002JD003359. <http://doi.wiley.com/10.1029/2002JD003359>
- Suzuki, K., T. Nakajima, A. Numaguti, T. Takemura, K. Kawamoto, and A. Higurashi, 2004: A study of the aerosol effect on a cloud field with simultaneous use of GCM modeling and satellite observation. *J. Atmos. Sci.*, **61**, 179–194, doi:10.1175/1520-0469(2004)061<0179:ASOTAE>2.0.CO;2.
- , T. Y. Nakajima, and G. L. Stephens, 2010: Particle growth and drop collection efficiency of warm clouds as inferred from joint CloudSat and MODIS observations. *J. Atmos. Sci.*, **67**, 3019–3032, doi:10.1175/2010JAS3463.1. <http://journals.ametsoc.org/doi/abs/10.1175/2010JAS3463.1>
- , J.-C. Golaz, and G. L. Stephens, 2013: Evaluating cloud tuning in a climate model with satellite observations. *Geophys. Res. Lett.*, **40**, 4464–4468, doi:10.1002/grl.50874. <http://doi.wiley.com/10.1002/grl.50874>.

Reviewers' Comments:

Reviewer #1:

Remarks to the Author:

The authors have made substantial and constructive changes to address the concerns raised in the review. The manuscript now more clearly presents its results even as it digs more deeply into the models response. The improvements presentation, the supplementary analysis, and the very constructive response to the reviews combine with my sense that the manuscript, as the first ever to look at aerosol responses using a fundamentally more advanced modelling framework, offers a very important contribution to our understanding of factors that influence modelled aerosol radiative forcing. Based on this I am happy to recommend it for publication in Nature communications.

Reviewer #2:

Remarks to the Author:

The authors have addressed my previous comments, but the answer to what microphysics scheme was used in their model simulations introduces new questions. They used the same single-moment cloud microphysics scheme (i.e., N_c is not predicted) and empirical aerosol-cloud interaction parameterization for both GCM and GCRM. Therefore, the differences between GCM and GCRM are only because of more resolved clouds, which is only a resolution problem. So, the introduction and discussion should be modified toward this standpoint. As I commented previously, the state-of-art GCMs use two-moment cloud microphysics with N_c explicitly predicted. This paper would be much stronger if the GCRM results are also compared with a state-of-art GCM using a two-moment cloud microphysics for aerosol-cloud interactions. If your single-moment GCM result is similar to that of the state-of-art GCM with a two-moment scheme, then you can actually make a point that improving model resolution to resolve clouds would be more critical than improving cloud microphysics parameterizations.

Another comment about single-moment scheme is that it would overestimate rain evaporation a lot compared with a more physical treatment such as two-moment and bin model (basically, the rain evaporation can not be physically represented with the one-moment treatment) (you may read some CRM studies in which different microphysics schemes are compared about this, for example, Li et al., 2009, *J. Atmos. Sci.*, 66, 3–21, and Wang et al., 2013, *J. Geophys. Res.*, 118, 5361–5379). This will produce a large uncertainty to your conclusion about the response of evaporation to aerosol variation. This caveat at least should be noted and discussed.

Reviewer #3:

Remarks to the Author:

The authors have done a comprehensive job responding to all three reviewers concerns and comments. The manuscripts reads much better than the previous version while still conveying the important scientific results. The added details add weight to the paper and will help readers better understand the importance and implications of the findings, as well as to devise follow-on studies. I recommend publication of the manuscript.

Reply to the Reviewer #1:

The original title of the manuscript: 'What is required for accurate global modelling of aerosol effects on cloudiness?', by Y. Sato et al. submitted to Nature Communications, whose manuscript number is NCOMMS-17-18141A.

Firstly, we should note that we modified Fig. 1e because the previous version was wrong. Even if the new figure has some difference from the previous version, the main proposal of this figure (λ was converted from positive or about 0 to negative with increasing precipitation) doesn't change.

Comment from reviewer:

The authors have made substantial and constructive changes to address the concerns raised in the review. The manuscript now more clearly presents its results even as it digs more deeply into the models response. The improvements presentation, the supplementary analysis, and the very constructive response to the reviews combine with my sense that the manuscript, as the first ever to look at aerosol responses using a fundamentally more advanced modelling framework, offers a very important contribution to our understanding of factors that influence modelled aerosol radiative forcing. Based on this I am happy to recommend it for publication in Nature communications.

We are grateful for the reviewer's efforts to read our manuscript and we are very thankful for the reviewer to admit the publication of our manuscript from Nature Communications.

Reply to the Reviewer #2:

The original title of the manuscript: 'What is required for accurate global modelling of aerosol effects on cloudiness?', by Y. Sato et al. submitted to Nature Communications, whose manuscript number is NCOMMS-17-18141A.

We are grateful for the reviewer's efforts to read our manuscript and give useful comments.

Firstly, we should note that we modified Fig. 1e because the previous version was wrong. At the same time, Fig. 1f was also modified to use same horizontal and vertical axis as the modified Fig. 1e. Even if the revised figure has some difference from the previous one, the main proposal by this figure (λ was converted from positive or about 0 to negative with increasing precipitation) doesn't change. So, we believe that the conclusion of this manuscript is not changed by the modification of Fig. 1e.

As well as the modification, we added discussions about the uncertainties in the one moment bulk microphysical model in our revised manuscript based on the reviewer's comments.

The modified parts are highlighted in red character in the revised manuscript.

Our answers written in black letters to the reviewer's comments, which are written in blue letters, are shown below. The Lxx means line number "xx"

We are very happy if the reviewer accepts our manuscript for publication from Nature Communications.

Comment:

C: The authors have addressed my previous comments, but the answer to what microphysics scheme was used in their model simulations introduces new questions. They used the same single-moment cloud microphysics scheme (i.e., Nc is not predicted) and empirical aerosol-cloud interaction parameterization for both GCM and

GCRM. Therefore, the differences between GCM and GCRM are only because of more resolved clouds, which is only a resolution problem. So, the introduction and discussion should be modified toward this standpoint. As I commented previously, the state-of-art GCMs use two-moment cloud microphysics with N_c explicitly predicted. This paper would be much stronger if the GCRM results are also compared with a state-of-art GCM using a two-moment cloud microphysics for aerosol-cloud interactions. If your single-moment GCM result is similar to that of the state-of-art GCM with a two-moment scheme, then you can actually make a point that improving model resolution to resolve clouds would be more critical than improving cloud microphysics parameterizations.

A: Thank you for your useful comment. As we described in the previous version of the manuscript (please see L214 of the previous manuscript and L217-218 of the revised manuscript), our GCM (MIROC) used double moment bulk microphysical scheme. The GCM explicitly predicted the number concentration of cloud (N_c). The GCM has been participated in model intercomparison studies (e.g., CMIP5, CFMIP, and so on), and it is now participating a latest model intercomparison of GCM (i.e., CMIP6). Thus, we can regard our GCM as one of the state-of-art GCMs. In our manuscript, we described our GCM as traditional one to emphasize the difference between the GCM and GCRM, but, our GCM is not conventional (traditional) one. Thus, the analyses of this study are the comparison between GCRM and state-of-art GCM.

We think that the reviewer regarded our GCM using one moment scheme because of the description of Line 224 of the previous version of the manuscript. In this part, we described that both GCM and GCRM used N_{ccn} predicted by SPRINTARS for N_c of Eq. (1) to implement aerosol effect. This is our mistake. The N_{ccn} was used only for GCRM. N_c of Eq. (1) is directly predicted in GCM. We modified this part. Please see L215-228 of the revised manuscript.

C: Another comment about single-moment scheme is that it would overestimate rain evaporation a lot compared with a more physical treatment such as two-moment and bin model (basically, the rain evaporation can not be physically represented with the one-moment treatment) (you may read some CRM studies in which different microphysics schemes are compared about this, for example, Li et al., 2009, J. Atmos. Sci., 66, 3–21, and Wang et al., 2013, J. Geophys. Res., 118, 5361–5379). This will

produce a large uncertainty to your conclusion about the response of evaporation to aerosol variation. This caveat at least should be noted and discussed.

A: Thank you for the useful comment. We absolutely understand the uncertainties of the one moment bulk schemes. As you indicated, the one moment bulk scheme has room for the improvement as reported by several previous studies that you showed. As well as the studies, several intercomparison studies using LES models suggested the large impacts of the cloud microphysical scheme on simulated clouds (e.g., VanZanten et al. 2011). Of course, we also investigated the sensitivity of the cloud microphysical scheme by ourselves (Sato et al. 2015).

Even if the one moment bulk scheme has the uncertainties, we believe that the results of this study are useful for our community because the simulation of GCRM coupled with aerosols transport model was conducted for one year for the first time.

We will conduct the same type of the simulation by using double moment bulk scheme in future when next-generation super computer is developed.

Based on your comment, we added some descriptions about the uncertainties of the one moment bulk scheme in the method of the revised manuscript. The references that you showed were added in the part. Please see from L229 to L235 of the revised manuscript.

References

- Sato, Y., S. Nishizawa, H. Yashiro, Y. Miyamoto, Y. Kajikawa, and H. Tomita, 2015: Impacts of cloud microphysics on trade wind cumulus: which cloud microphysics processes contribute to the diversity in a large eddy simulation? *Prog. Earth Planet. Sci.*, **2**, 23, doi:10.1186/s40645-015-0053-6.
- VanZanten, M. C., and Coauthors, 2011: Controls on precipitation and cloudiness in simulations of trade-wind cumulus as observed during RICO. *J. Adv. Model. Earth Syst.*, **3**, M06001, doi:10.1029/2011MS000056.

Reply to the Reviewer #3:

The original title of the manuscript: 'What is required for accurate global modelling of aerosol effects on cloudiness?', by Y. Sato et al. submitted to Nature Communications, whose manuscript number is NCOMMS-17-18141A.

Comment from reviewer: The authors have done a comprehensive job responding to all three reviewers concerns and comments. The manuscripts reads much better than the previous version while still conveying the important scientific results. The added details add weight to the paper and will help readers better understand the importance and implications of the findings, as well as to devise follow-on studies. I recommend publication of the manuscript.

We are grateful for the reviewer's efforts to read our manuscript and we are very thankful for the reviewer to admit the publication of our manuscript from Nature Communications.